# Examining the Nonlinear Effects of Residential and Workplace-built Environments on Active Travel in Short-Distance: A Random Forest Approach

**DOI:** 10.3390/ijerph20031969

**Published:** 2023-01-20

**Authors:** Liang Guo, Shuo Yang, Yuqing Peng, Man Yuan

**Affiliations:** 1School of Architecture & Urban Planning, Huazhong University of Science and Technology, Wuhan 430000, China; 2The Key Laboratory of Urban Simulation for Ministry of Natural Resources, Wuhan 430000, China

**Keywords:** nonlinear effects, travel mode choices, short-distance commuting, random forest approach

## Abstract

Environmental pollution and health problems caused by the excessive use of motor vehicles have received widespread attention from all over the world. Currently, research lacks attention to the nonlinear effects of the built environment on short-distance active travel choices. It is important to understand these non-linear correlations, because it would be more feasible and necessary to promote a shift from car users to walking and cycling mode choices over short commuting distances. A random forest model was used to analyze the nonlinear effects of residents’ social characteristics and the built environment of their homes and workplaces on their choice of walking and cycling. The results show that the built environment has a greater impact on short-distance active travel than the socio-demographics attributes. Residential and workplace-built environments have equal importance and they have significant non-linear effects on both short-distance walking and cycling. The nonlinear effects of the built environment on walking and cycling differed significantly, and the study specifically revealed these effects.

## 1. Introduction

As the number of cars continues to increase annually, environmental pollution and health problems caused by excessive car use remain a global concern. Academics believe that replacing car travel with active travel can reduce carbon emissions [1,2] and benefit people’s health [2,3]. The possibility of using active modes such as walking and cycling as alternatives to driving was first explored in European countries [4], and it was discovered to have great potential [3,5,6]. Particularly during the COVID-19 pandemic, the importance of bicycles as a mode of transportation was reinforced, as they provided an efficient means of transportation while limiting physical proximity. The pandemic has promoted increased efforts to encourage cycling, particularly in large cities in Europe, US, and Australia [7]. On one hand, travel distance is a key determinant of commuting mode choice, and it is often more feasible to encourage residents to walk [8] and cycle [9] for short distances due to physical constraints. On the other hand, research has shown that a large number of residents use cars for short distances [5]. The preference for motorized travel in short trips compared to active travel is an important cause of deterioration in the traffic trip structure and thus traffic and public health problems [10,11]. Therefore, it is feasible, necessary and has significant environmental benefits to replace some car trips for short trips with active travel by walking or cycling [10,12,13].

How can we promote people’s choices of active travel? More and more studies have focused on looking for reasons why people use their cars for short distance travels [14,15,16]. They have explored the reasons for car travel choices in terms of residents’ attitudes, preferences, socio-demographic attributes, and urban built environments. Although most studies have concluded that active travel choices can be promoted by improving the built environment [17,18,19], the relative importance between the above elements in short distance travel remains inconclusive. Many studies have confirmed that nonlinear associations between variables are common in travel behavior [20,21,22], but nonlinear associations are still ignored in studies of short distance travel. In addition, in most studies, walking and cycling were considered separately, and rarely both were considered together. More importantly, a fine portrayal and comparative analysis of the influencing elements and mechanisms of walking and cycling is missing. The gaps in these studies has hindered the refinement of urban planning and transportation policies. Especially in view of the fact that many policies intended to promote active travel were introduced, the continued excessive use of cars for short distance travels remains.

To fill these gaps, this study used a random forest approach to capture the complex relationship between active travel mode choice and built environment variables in short-distance travel. We used a dataset with a larger sample size (20,008) to avoid overfitting. It makes a dual contribution to the established literature: first, it focuses on the nonlinear and threshold effects of active travel mode choice in the built environment and short-distance commuting behavior, and explores the similarities and differences between walking and cycling as influenced by the environment. Second, by revealing the different mechanisms linking the built environment of residence, workplace and active travel, it can provide effective suggestions for appropriate environmental interventions and encouraging active travel modes.

This manuscript is divided into five sections. Except for this section, Section 2 provides a literature review, identifying research gaps. Section 3 presents the data, variables, and the random forest approach. Section 4 provides details of the results obtained, and Section 5 summarizes the main findings and discusses their implications for planning practice. Section 6 describes the main conclusions of this study.

## 2. Literature Review

Over the past decades, research on residents’ travel mode choices has expanded rapidly and yielded insightful results. Extensive research has shown that travel mode choice is influenced by multiple factors such as the built environment and socio-economic attributes [23,24,25]. The influence of personal characteristics on travel choices is mainly reflected in the variability in travel mode choices among people of different age, gender, income, education, and family structure. Research by Dėdelė et al. [26] showed that male, young and employed people are more likely to choose to travel by car. A study by Cheng et al. [27] pointed out that females were more likely to choose public transport than males. Li et al. [28] concluded that car use was negatively associated with increasing age. A study by van den Berg et al. [29] showed that people with higher education were more likely to use public transportation.

Many studies have concluded that the built environment plays a more critical role in travel mode choice. Most studies focusing on the built environment showed that “5D” elements of the built environment (density, diversity, design, destination accessibility and distance to transit service) are closely related to travel mode choice [23,30,31]. In general, people living in high-density and high-mix areas are more likely to choose to walk and cycle because they have more options for travel goals and are therefore often closer to their destinations [19,20]. Pedestrian and bike-oriented neighborhood design is more conducive to active travel, such as higher road density, intersection density, and street connectivity [32,33]. These studies showed that the factors influencing travel mode choice are multidimensional. In this context, it is particularly important to identify the key variables for travel mode choice [32,34]. It is crucial for planners to know the priority level of the determinants of travel mode choice in order to develop the right interventions [35,36]. In recent years, a limited number of studies used machine learning methods to assess the relative importance of these elements [32,33,37]. They found that the impact of the built environment was more important than other variables. In addition, early studies mainly counted the built environment indicators within a certain area centered on residence, but in recent years, some studies have started to consider the built environment of residence and workplace together, and found that the environmental indicators of the place of work also have a non-negligible influence on the travel mode choice [22,38,39].

Overall, these studies have focused on the effects of the built environment on the choice of full distance travel mode. However, the elements and relative importance that influence the short-distance travel choices remain unattended. Understanding the elements and relative importance that influence short-distance travel choices is meaningful to planners. Many Chinese cities are currently devoted to creating 15 min pedestrian-scale neighborhoods for residents to reduce residents’ daily travel distance and promote active travel. These policies are indeed beneficial in reducing long distance travel for residents. However, there are still some people who choose to travel short distances by car. What elements are more important to encourage short distance active travel? What factors should be prioritized in order to promote short distance walking and cycling, respectively? This study fills in the gaps by quantifying the extent to which the built environment affects short-distance active travel and comparing the extent to which multiple built environment dimensions of residence and workplace contribute differently to cycling and walking. The results can help planners prioritize improvements and support the design of more targeted planning strategies, because there is more potential to achieve a shift from car travel to active travel in short distance travel [10,12,13].

Alternatively, while established studies have explored the association between the built environment and active travel, many of them presuppose linear [40,41,42] or log-linear [43] relationships between variables. In recent years, many studies have confirmed that nonlinear associations between variables are common in travel behavior [20,21,22,44], but nonlinear associations are still ignored in studies of short-distance travel. Indeed, accurately identifying this nonlinear relationship is critical because the threshold effect implies that land-use policies to promote active travel by changing built environment variables may only be effective within a range of values [21,22], and the cost of policy implementation can be significantly reduced by identifying a reasonable range and threshold [45]. For example, Ding et al. [22] showed that bus stop density had a limited effect on the probability of auto commuting, but above a threshold of 0.2 persons per acre, its negative correlation increased sharply. In addition, population density around residential areas is negatively correlated with car mode choice for commuting, but 35 persons per acre is the threshold of impact, beyond which the marginal impact diminishes. That is, if car commuting is reduced by changing density, the increase in density is not unlimited and the threshold should be 35 persons per acre.

It is important to reveal this nonlinear relationship in China, where the scale and speed of change in the built environment exceeds that of North America or Europe [46,47]. The experience of Europe and the United States may not be applicable. In addition, in China, many cities are attempting to promote walking or cycling through planning policies. For example, the design guidelines for pedestrian-oriented streets in Wuhan recommend a road network spacing of no more than 200 m and a road network density greater than 8 km/km^2^. Are these values correct? Limited evidence is provided in the literature, and the planning orientation is different in different areas. For example, in the older parts of Wuhan, planning policies tend to encourage people to commute on foot, while in some of the newer, faster-paced urban areas, policies encourage people to commute by bike. This shows that walking and cycling still need to be treated differently as active modes of travel as well. This study uses a random forest approach to identify the non-linear effects of the built environment of the residence and workplace on two active travel modes, walking and cycling, and to assess the relative importance of different factors. This study helps to develop precise urban planning and transportation policies to promote active travel as an alternative to short-distance motorized travel, which will greatly improve the urban transportation travel structure, urban environment, and individual well-being.

## 3. Materials and Methods

### 3.1. Study Site

Wuhan was selected as a case study (Figure 1) as it has the highest population size among the fourteen megacities in China, and is a center for scientific research and education in central China. Wuhan’s motor vehicle fleet grew rapidly after 2000, with the total number of motor vehicles exceeding four million by 2021. Despite the increased investment in transportation facilities in recent years, traffic congestion in Wuhan has been difficult to improve. According to the city’s future plan, the projected resident population in 2035 will be 16.6 million, and the travel needs of future residents will be considerably larger.

### 3.2. Data Sources

#### 3.2.1. Resident Travel Survey

The travel data in this study came from the fourth travel survey of residents in 2020 conducted by the Wuhan Institute of Transportation Development Strategy. Surveys have been conducted every decade to determine the basic characteristics of daily movements of the population. A structured Family Interview Questionnaire was used to collect daily travel information from residents, such as the start and end points of daily travel, purpose of travel, selected mode of travel, and social and economic characteristics of families and individuals, including age, gender, personal education level, employment status, household registration, family income, family size, and number of children. The survey was conducted in Wuhan with a 0.5% sample rate (Table 1). Researchers conducted random household surveys using the WeChat app. A total of 43,660 people’s travel information were collected over a two-month period. The data were further processed, missing information was removed, and 30,174 samples were obtained. The spatial distribution of the samples is shown in Figure 1.

#### 3.2.2. Built Environment Data

The Wuhan Urban GIS database provides measured data for variables in the built environment (Table 2). We collected the built environment characteristics within 800 m of respondents’ residences and destinations. Among them, dwelling location characteristics were expressed in terms of the Euclidean distance of the residence from the urban center and the nearest cluster center. The cluster center is identified by Wuhan LBS data. (The results are from another study by the authors [48].) The mixed entropy index of land use considers six types of land: residential, commercial, educational, industrial, public services, and green space. Data on the resident and employed populations were obtained from the Wuhan census.

### 3.3. Methods

#### 3.3.1. Identification for Short Distance Travel

Determining the distance thresholds for short trips is critical to this study. We determined the short-distance travel threshold based on a combination of the established literature and the results of Wuhan travel survey data. First, the range of “short commute” distance thresholds varies by region and topic. Mackett interviewed 377 people in the United Kingdom who traveled short distances by car, setting the short-distance threshold at 8 km [5]. Nazelle et al. [11] defined the short-distance travel threshold at 3 miles (4.8 km) in a study that analyzed the probability of US residents switching from small cars to other modes of transportation. Beckx et al. [49] defined the short distance threshold at 8 km in their study analyzing the potential of the Dutch to replace car trips with active trips. When assessing the potential carbon savings of walking and cycling instead of driving short distances in Wales, Neves defined short trips as less than 3 miles (5 km) in length [6]. Similar research conducted in the cities of China consider 5 km as a short distance [50]. Second, the commuter travel distance of a single sample was calculated using the Baidu Map path-planning API based on the individual commuter mode obtained from the survey by vectorizing the spatial coding of the OD points of commuter travel obtained from the survey. According to the results, the average distance of walking and cycling in Wuhan is 1.25 km. Moreover, 96.97% of walking or cycling commuting distance is within 5 km (Table 3). Therefore, in order to effectively examine short-distance active travel, a 5 km threshold is chosen as the definition of short-distance travel in this paper for the study.

A total of 20,008 short-distance travel samples with travel distance less than 5 km were screened, accounting for 66.31% of the total sample (Figure 2).

#### 3.3.2. Variable Descriptive Statistics

Further descriptive statistics were conducted on the screened short distance travel sample and the results are shown in Table 4. Among them, we counted the built environment of the respondents’ residence and destination separately.

#### 3.3.3. Modeling Method

The random forest (RF) model was used in this study to uncover the complex relationship between multidimensional elements and commuting carbon emissions. Ho [51] proposed RF, which is based on the integration method of decision trees, and optimizes model fitting and prediction by assembling a large number of individual decision trees [52]. It extracts multiple samples from the original samples using the bootstrap resampling method, and builds a decision tree model for each bootstrap sample. The split in each tree lasts until the tree reaches its maximum depth. The forecasts of multiple decision trees are then combined, the final forecast is obtained through voting, and the final result is obtained by averaging the forecasts of all individual trees. The working route is shown in Figure 3.

Numerous theoretical and empirical studies have shown that RF is highly predictive, robust to outliers and noise, and not prone to overfitting [53,54]. Most importantly, it directly expresses the true relationship between variables rather than assuming a specific parameter relationship between the self and dependent variables, as in traditional linear regression [55]. The RF algorithm can handle missing values and maintain accuracy if some data are lost. In addition, the RF algorithm performs better on a large sample dataset and is less sensitive to outliers than another popular machine learning algorithm, GBDT. In addition, RF is better than GBDT for classification tasks.

For model calibration, three parameters must be considered: the total number of trees n (forest size), the number of split variables m, and the maximum tree depth d [33]. Alternatively, it can calculate the relative importance of a single variable in a variable dataset and create partial dependency plots to show the relationship between the independent and dependent variables. Furthermore, in recent years, some researchers have questioned whether the higher fit of RF compared to linear forest is due to overfitting [56]. This study used a dataset with a larger sample size (20,008) to avoid overfitting. Finally, we use RF to predict the probability of walking (Model 1) and cycling (Model 2) mode separately.

## 4. Results

After referring to the relevant literature [20,33,57], and after several tests, this study used Python3.8 to build a RF model, with 20% of the samples randomly selected as the test set and 80% as the training set, in order to obtain more reliable model results. The parameters were optimized in two steps using Bayesian estimation with the Hyperopt algorithm: (1) determine the best decision tree scale n, and (2) determine the best split variable number m and tree depth d. Finally, after 300 iterative optimizations, optimal model parameters are determined. The results show that the predictive performance of model 1 can be neglected, and more time is consumed after more than 35 trees, while the number of trees for model 2 is 141; therefore, the number n of trees of model 1 was set to 35. The number n of trees of model 2 was set to 141. The prediction performance of model 1 is best when the maximum tree depth d is 46 and the split variable m is 6. While the prediction performance of model 2 is best when the maximum tree depth d is 24 and the split variable m is 6. The final R^2^ of model 1 was 0.77, and the final R^2^ of model 2 was 0.80 (Table 5) (the R^2^ of the logistic regression model with the same independent variable data set was 0.61 (Table A1) and 0.65 (Table A2). The final models were used to quantify the importance of each built environment variable and draw partial dependency plots.

### 4.1. Relative Contributions of Independent Variables

Table 6 illustrates the relative importance of the independent variables in predicting the individuals’ choice of active travel. The sum of the relative importance of all independent variables was 100%, and the ranking was based on the magnitude of the relative importance. Overall, the total contribution of the eight personal attributes to the influence role of walking and cycling was 16.78% and 14.70%, respectively. The total contribution of the ten built environments of residence to the influence role of walking and cycling was 41.13% and 42.92%, respectively, and the total contribution of the ten workplace-built environments to the influence of walking and cycling was 42.09% and 42.38%, respectively. This indicates that built environment factors of work and home are equally important in predicting active travel mode choice and are more important than personal socio-demographic characteristics of residents, which supports previous research that land use variables have a greater impact on travel than demographics [19]. This result suggests that optimizing the built environment is important for active travel, and that the residential built environment and workplace-built environment should receive equal attention. The eight individual socioeconomic attribute variables contribute in almost equal ranking to walking and cycling mode choice, indicating that the relevant personal attribute elements had the similar influence on both active travel choices. Age had the strongest influence on walking and cycling, followed by education level and household economic factors, whereas gender and car ownership has little influence on active travel patterns, with similar results found in studies by Cheng et al. [37]. The results also show that residential built environment factors have a slightly greater influence on the choice of cycling mode than on walking mode.

More specifically, the contribution of residential-built environment elements to walking mode choice was 41.13%, which was slightly less than the influence of workplace-built environment elements to cycling mode choice (42.92%). For walking, the top three most influential factors were: distance to the nearest cluster center (5.56%), distance to the city center (4.51%), and job density (4.39%), which is consistent with the strongest relationship between accessibility to regional centers and travel behavior in several studies [58,59]. The three most influential factors on cycling were: job density (4.83%), land-use diversity (4.68%), and road network density (4.67%), which is consistent with previous research suggesting that compact development leads to less driving [60]. This result suggests that for walking, residential location has the greatest impact, as residents living in urban centers tend to travel shorter distances and they are more likely to walk. For cycling, road density and land use mix are more important than district location, suggesting that cycling responds more significantly to neighborhood design and land use characteristics. Therefore, to encourage cycling, these two points need to be prioritized. The job density near residential plots is important for both walking and cycling, which demonstrates a relative balance of jobs and housing is more conducive to promoting active travel.

The contribution of workplace-built environment factors to walking and cycling were nearly identical, at 42.09% and 42.38%, respectively. All ten built environment variables selected for the study explain active travel to a high degree, but the ranking of the degree of influence of individual elements varies greatly. The top three most influential elements for walking were: distance to the city center (4.77%), distance to the center of the nearest cluster center (4.67%), and road network density (4.65%). Moreover, the three elements with the greatest degree of influence on cycling were: population density (5.04%), distance to urban centers (4.71%), and job density (4.66%). The results show that the population and job density of the workplace has a greater impact on cycling, while the workplace location and road network density have a greater impact on walking, which leads to a different conclusion from the residential-built environment. The conclusion suggests that, while focusing on jobs and housing balance, planning policies should be differentiated for job centers and residential centers.

### 4.2. Nonlinear Effects of Residential Built Environment Variables

Figure 4 and Figure 5 depict the threshold effect of residential built environment elements on the short distance active travel mode choices. Overall, most residential built environment elements have non-linear effects on short-distance active travel. This confirms the nonlinear relationship between the residential built environment and short distance travel behavior. Figure 4 and Figure 5 show the specific nonlinear effects of typical residential built environment elements on short distance walking and cycling.

The effect of the distance of residence from the city center on the mode of cycling trips was not markedly significant after 15 km, but it began to show a significant facilitating effect on walking. The threshold point for a mode shift in the walking mode is 15 km, whereas the threshold for the turning point in cycling mode is smaller (in the 0–8 km range, the further the residence is from the city center, the less likely residents are to choose cycling; within 8–15 km, the negative effect reverses to a positive effect). The distance to the nearest cluster center also has different effects on walking and cycling. Although the thresholds of impact are all at 4 km, this distance is positively correlated with the probability of walking when the distance to the nearest cluster center is less than 4 km, and the marginal effect slows down beyond 4 km; when the distance to the nearest cluster center is less than 4 km, this distance is negatively correlated with the probability of cycling, and then becomes positively correlated. This suggests that the effects of location on walking and cycling are different, and that policies to encourage walking and cycling should be imposed on communities in different locations.

For land use characteristics, land use entropy index and land use intensity have non-linear effects on both short distance walking and cycling. However, the non-linear effects of land use characteristics on short distance walking and cycling differ. The land use entropy index has opposite effects on walking and cycling, specifically, as land use entropy index exceeds 0.65 and continues to increase, the probability of walking gradually decreases, while the probability of cycling gradually increases. The effects of land use intensity on walking and cycling are similar, with the probability of both walking and cycling decreasing and then increasing as land use intensity rises. The difference is that the threshold for the impact of land use intensity on walking is at 2.00 and for cycling at 3.20. This suggests that different land use controlling indicators should be chosen for pedestrian and cycling oriented neighborhoods.

Job density of residence is important for both walking and cycling, but job density has very different non-linear effects on the two active travel modes. Job density has a significant positive effect on walking, especially when the job density exceeds 12,500 jobs/km^2^, the positive effect becomes more significant. However, the job density has the opposite effect on cycling. As the job density increases, the probability of cycling decreases significantly, and when it exceeds 10,000 jobs/km^2^, the negative effect diminishes and becomes a weak positive effect. Road network density has a significant positive effect on walking, while there is a significant cut-off point for cycling. Before the road network density reaches 5.50 km/km^2^, the road network density and cycling are negatively correlated, when the road network density exceeds 5.50 km/km^2^, the negative correlation suddenly changes to a significant positive correlation. This result proves that, in general, the rise in road network density facilitates active travel and that the minimum value of road network density should be 5.50 km/km^2^, and should be further improved.

### 4.3. Nonlinear Effects of Workplace- Built Environment Variables

Figure 6 and Figure 7 show the threshold effects of workplace-built environment elements on the choice of an active travel modes in short distance travels. Overall, most workplace- built environment elements also have nonlinear effects on short-distance active travel. Figure 6 and Figure 7 show the specific nonlinear effects of typical workplace-built environment elements on short distance walking and cycling. These effects differ from the relationship between the residential built environment and short-distance active travel.

The distances from workplace to urban and nearest cluster center have significant and varied effects on short distance active travel. Regarding the walking mode of transportation, the closer the workplace is to the city center, the more likely it is to be chosen when the distance from the urban center is between 0 and 8 km. After a distance of more than 8 km, the further the workplace is from the city center, the more likely people are to choose to walk. The effect of distance from the workplace to the nearest cluster center on the choice of walking showed a significant positive effect. However, this effect was tapered after more than 27 km. For the cycling travel mode, both the distance from the city center and the distance from the cluster center show a positive correlation with the choice of cycling, but the degree of influence is greater for distance from the city center, and the influence range is within 32 km, which is farther than the influence range of the proximity to the cluster center (27 km).

The road network density at the workplace affects walking and cycling choices in completely opposite ways, with a road network density of 5 km/km^2^ being an important turning point. A road network density below 5 km/km^2^ promotes the choice of walking mode and discourages the choice of cycling mode, whereas a road network density above 5 km/km^2^ discourages the choice of walking mode and promotes the choice of cycling mode.

There are also large differences in the way workplace land use entropy index affect walking and cycling. At a land use entropy index less than 0.62, the increase in diversity has a slight contribution to walking travel mode choice and a greater inhibitory effect on cycling travel mode choice. At a land use entropy index greater than 0.62, the land use entropy index sharply decreases the promotion of walking and greatly promotes cycling. The average value of land use entropy index in Wuhan has already reached 0.65, which is higher than most developed countries in Europe and the United States; therefore, for a city like Wuhan, a further increase in land use mix is less effective in promoting short distance active travel.

In terms of transit accessibility, the farther the distance from the nearest public transportation station, the greater the possibility of choosing walking and cycling. This proves that as transit accessibility decreases, active travel is a substitute for transit travel. However, the number of public transit stations on the choice of walking and cycling mode is with an increase in number, showing the effect of first inhibiting and then promoting. For walking, the threshold point for the number of public transit stations is at 20, while for cycling, the threshold point for the number of public transit stations is at 25. Above these thresholds, the possibility of both walking and biking rises as the number of transit stops increases.

## 5. Discussion

This study examined the nonlinear effects of workplace and residential built environments on short-distance active travel using a random forest approach. First, this study found that the built environment of residence and work had a relatively consistent effect on walking and cycling. This differs from the conclusions reached by studies in Western countries [22]. This paradox is largely related to the vast differences in residential choices between Western and Chinese residents. Cities such as the U.S. have a large number of residential areas in the suburbs with low densities and a single type of land use, while Chinese residents tend to purchase homes close to the city center and sites near residential areas also tend to have a higher distribution of jobs. Thus, residential locations in China’s high-density cities also influence residents’ active travel choices as much as their workplace locations. This suggests that place of residence and place of work should be given equal weight in policy development. Second, the study also found that the built environment has a greater impact on short-distance active travel than individual socioeconomic attributes. Consistent with the findings of previous studies [19,22,33], this study further demonstrates the importance of improving the built environment to promote short-distance walking and cycling as an alternative to car travel. In addition, the present study differs from the specific thresholds derived from studies on Western countries. They are mainly reflected in the elements of population density index, road network density index, and public transportation station index. On the whole, the indicator values in this paper are slightly higher than the research results in Europe and the United States, which may be due to the differences in the feedback effect of residents on the built environment in different countries, and is also related to the higher population density and the density of public transportation facility sites in Wuhan city is related to the overall higher density of public transportation facilities. In addition, the population density, land use mix and point density of transportation facilities in the central city of Wuhan have reached or even exceeded the optimal range for promoting active travel, and active travel can only be promoted through more microscopic environmental improvements, such as the improvement of bicycle lanes and increased street greening. On the other hand, the areas outside the third ring road of Wuhan are underdeveloped, so active travel can still be promoted by improving the land use mix and transportation facility sites.

More importantly, the built environment of the workplace and residence has a different effect on walking and cycling. Specifically, there are two thresholds for the “distance to the city center” indicator for both walking and cycling, ranging from 0 to 8 km, the closer to the city center, the more likely it is that both modes of travel will be chosen. In the 8–32 km distance range, the closer it is to the city center, the less likely it is to choose active travel. Beyond 32 km, the influence of the city center became very weak. Whereas the distance of residence and work from the city center act in the same way and threshold effect on cycling, there is a difference for walking, i.e., in the distance range of 0–13 km, the farther the residence from the city center, the less likely it is to choose walking, and after 13 km the opposite result is shown. It can be seen that the distance of residence from the city center has a more distant effect on the range of walking trips. The land-use entropy index had a minor effect on walking until 0.65, but a significant negative correlation with cycling, whereas the land-use mixture index had a minor effect on cycling after 0.65, but a significant negative correlation with walking. This suggests that land use diversity acts in different ways for walking and cycling. This differs from previous studies that concluded that land use diversity facilitates active travel [61,62]. This suggests that different land use patterns should be used for cycling and pedestrian-oriented areas, rather than generalizing and using the same land use indicators. In the older parts of Wuhan, planning policies tend to encourage people to commute on foot, while in some of the newer, faster-paced urban areas, policies encourage people to commute by bike. Different regions should adopt different land use indicators based on the findings of the empirical study.

Population density, both at residence and workplace, showed a significant positive correlation with walking trips. However, there is a threshold for the effect on cycling trips, that is, from 0 to 2500 persons/km^2^, which shows a negative correlation, and only after 2500 persons/km^2^ has a positive correlation. It demonstrated that a higher population density can be an advantage for the choice of active travel modes. This is similar to established studies. The road network density of residence and workplace can effectively promote the choice of cycling travel mode after reaching 5.5 km/km^2^, but has a significant inhibitory effect on walking, probably because the higher road network density makes the traffic more complicated, the danger of walking travel increases, and the urban environment is not conducive to walking commuting. This suggests that for pedestrian-oriented neighborhood designs, the density of the street network is not better, but rather the safety and quality of the pedestrian space should be improved to promote active travel [18].

The results of this study suggest that there are differences and contradictions in the extent and modality of the impact of the built environment on residents’ choices to walk or bike and that decisions need to be made in the context of the region’s own development goals. For example, in many Chinese cities, the creation of a 15 min walkable community requires the configuration of public transit stations, amenities, and other facilities to encourage walking. More importantly, revealing the range of thresholds for active travel in residential and workplace-built environments can provide fine-grained guidance for environmental construction.

This study has some limitations that can be a direction for future in-depth studies. First, the setting of short commute thresholds may differ for walking and cycling, which requires further research into the range of distances that residents are willing to walk or cycle to work, which may vary between cities. In addition, it is unclear whether the results of this study are consistent because Wuhan is dominated by flat terrain and no relevant studies are available for cities with large slopes.

## 6. Conclusions

To encourage active travel choices for short commutes, reduce carbon emissions from cars, and promote the health of residents, this study focused on the mechanisms by which the built environment at home and work affects the two main active travel modes of walking and cycling. In this study, a random forest model was used because there was no need to determine the parametric relationships between variables in advance, and a comparison with traditional logistics regression showed that random forests are more effective at revealing complex relationships. The main findings are as follows: (1) overall, the built environment is more important than individual socio-demographic attributes in influencing both types of active travel. (2) Residential and workplace-built environments are equally important in influencing active travel. (3) Most built environment factors have non-linear effects on the two active travel modes, but there are large differences in the way they affect walking and cycling.

Based on the above, the findings of this study are as follows. Built environment characteristics such as location, density, and land use entropy index are closely related to active travel. Therefore, these factors should be given priority in the ongoing planning and construction of territorial spatial planning to promote active travel. Since the place of residence and the place of work are equally important, community living area and business center area planning should receive equal attention. However, for cycling-oriented and pedestrian-oriented areas, differentiated planning policies should be developed to promote active travel. In addition, in recent years, Wuhan has been committed to improving spatial quality to promote active travel for residents to reduce their travel CO_2_ emissions and promote the health of the residents after the COVID-19 pandemic. The results of our empirical study can provide precise guidance for specific planning efforts.

## Figures and Tables

**Figure 1 ijerph-20-01969-f001:**
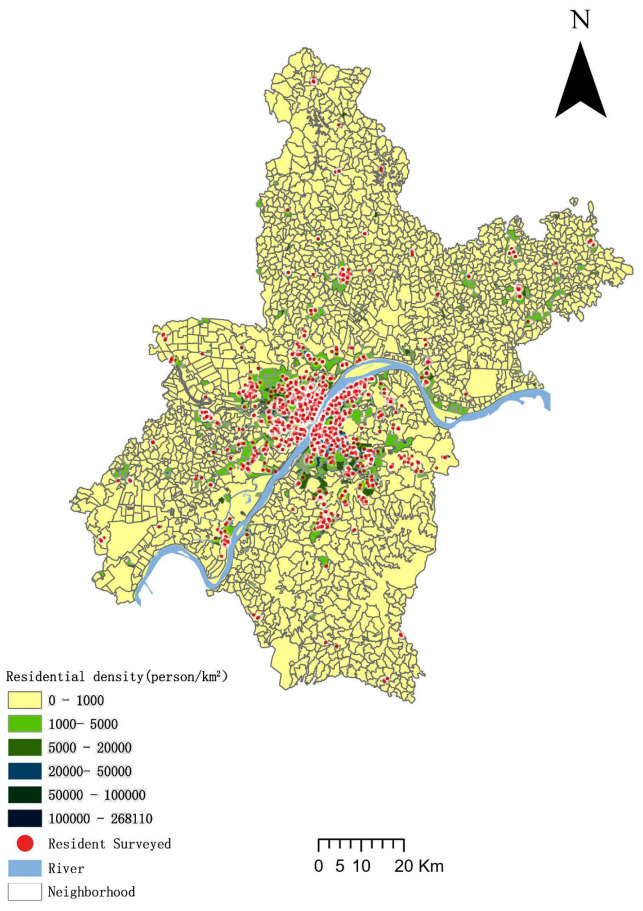
Spatial distribution of residents surveyed.

**Figure 2 ijerph-20-01969-f002:**
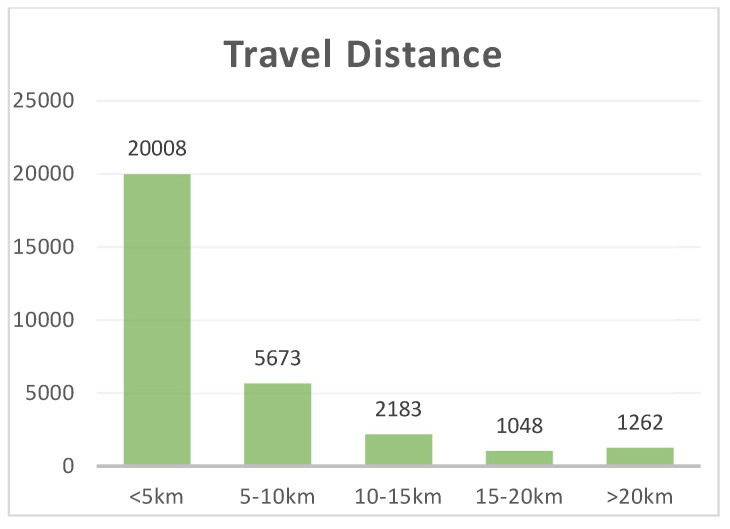
Wuhan commuting travel distance distribution.

**Figure 3 ijerph-20-01969-f003:**
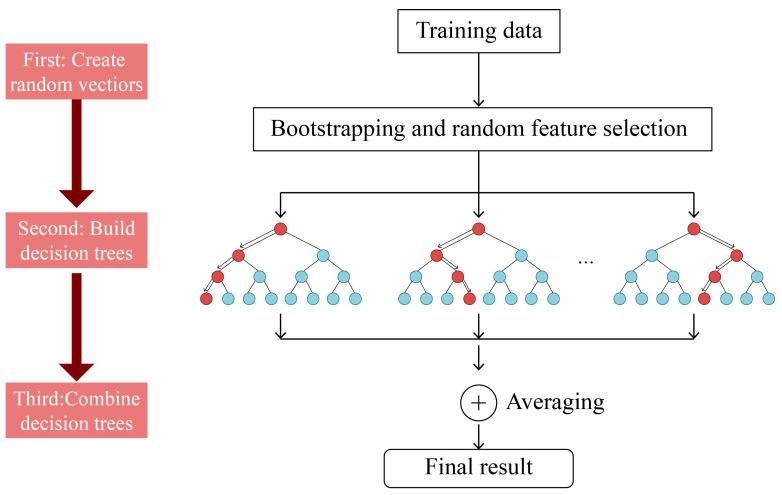
Working route of random forest algorithm.

**Figure 4 ijerph-20-01969-f004:**
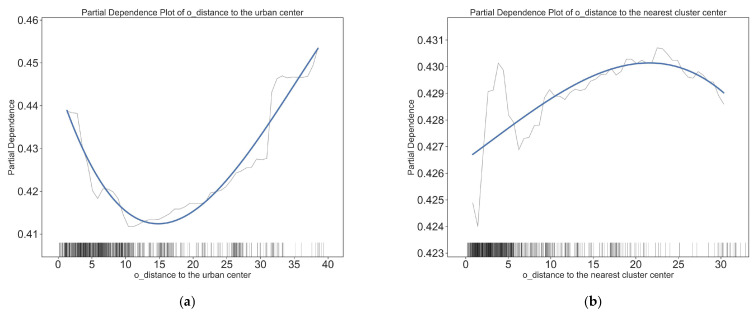
The effects of residential (O point) built environment variables on the choice to walk. (**a**) Impact of distance to the urban center on walking choices; (**b**) impact of distance to the nearest cluster center on walking choices; (**c**) impact of land use entropy index on walking choices; (**d**) impact of land use intensity on walking choices; (**e**) impact of job density on walking choices; (**f**) impact of road network density on walking choices.

**Figure 5 ijerph-20-01969-f005:**
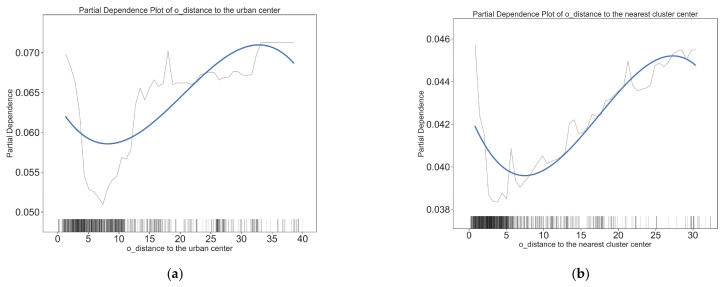
The effects of residential (O point) built environment variables on the choice to cycle. (**a**) Impact of distance to the urban center on cycling choices; (**b**) impact of distance to the nearest cluster center on cycling choices; (**c**) impact of land use entropy index on cycling choices; (**d**) impact of land use intensity on cycling choices; (**e**) impact of job density on cycling choices; (**f**) impact of road network density on cycling choices.

**Figure 6 ijerph-20-01969-f006:**
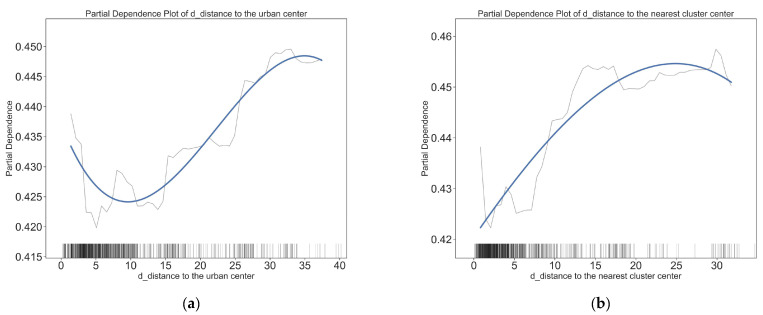
The effects of workplace-built (D points) environment variables on the choice to walk. (**a**) Impact of distance to the urban center on walking choices; (**b**) impact of distance to the nearest cluster center on walking choices; (**c**) impact of road network density on walking choices; (**d**) impact of land use entropy index on walking choices; (**e**) impact of distance to the nearest public transit station on walking choices; (**f**) impact of public transit station number on walking choices.

**Figure 7 ijerph-20-01969-f007:**
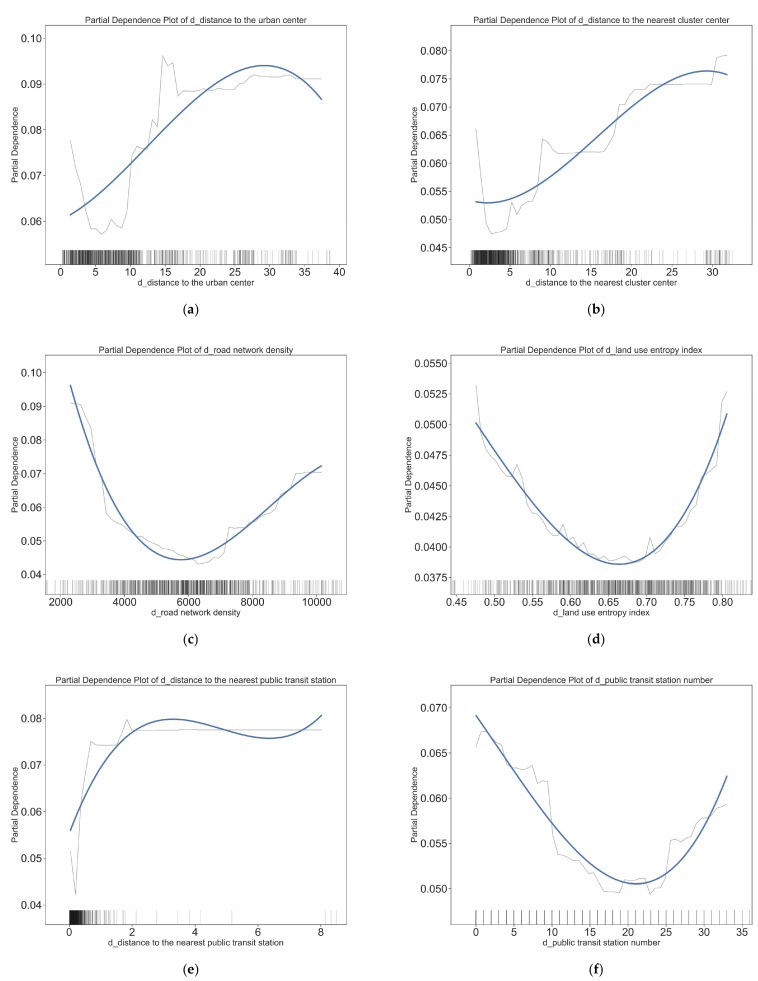
The effects of workplace-built (D point) environment variables on the choice to cycle. (**a**) Impact of distance to the urban center on cycling choices; (**b**) impact of distance to the nearest cluster center on cycling choices; (**c**) impact of road network density on cycling choices; (**d**) impact of land use entropy on cycling choices; (**e**) impact of distance to the nearest public transit station on cycling choices; (**f**) impact of public transit station number on cycling choices.

**Table 1 ijerph-20-01969-t001:** Basic information of data collection of four trips of residents in Wuhan.

Year	Zone	Resident Population and Scale	Survey Population and Household Size	Sample Rate
1987	Main urban area	330,000 people 40,000 households	—	—
1998	Main urban area	3,810,000 people 150,000 households	76,000 people 24,000 households	2.0%
2008	City area	870,000 people 200,000 households	120,000 people 38,000 households	1.5%
2020	City area	12,320,000 people 4080,000 households	43,660 people 15,000 households	0.5%

**Table 2 ijerph-20-01969-t002:** Built environment information table.

Variables			Description
Build environment	Location	Distance to city center	Distance from Wuhan city center—Hankou (in kilometers)
		Distance to the nearest cluster center	Distance to the center of the nearest sub-Cluster
	Transit accessibility	Distance to the nearest public transit station	Distance from the respondent’s residence and destination to the nearest bus stop (including subway and surface bus)
		Number of public transit stops	Number of public transit stops within 800 m buffer zone of respondents’ residence and destination
	Density	Population density	Residential density within 800 m buffer zone of respondents’ residential and destinations
		Job density	Job density within 800 m buffer zone of respondents’ residential and destinations
		Land development intensity	Land use intensity within 800 m buffer zone of respondents’ residential and destinations
	Design	Number of intersections	Number of intersections within 800 m buffer zone of respondents’ residential and destinations
	Diversity	Land use mixed entropy index	Land use diversity index within 800 m buffer zone of respondents’ residential and destinations

**Table 3 ijerph-20-01969-t003:** Active travel distance distribution.

Travel Distance	Number of Samples	Proportion
<5 km	9225	96.97%
5–10 km	271	2.85%
10–15 km	8	0.08%
15–20 km	3	0.03%
>20 km	6	0.06%

**Table 4 ijerph-20-01969-t004:** Variable statistics table.

Variables				Description		Mean.	Std.	Min	Max
Dependent variable	Travel mode		Walking	Whether the respondent commutes on foot, yes = 1, no = 0		0.42	0.49	0.00	1.00
			Cycling	Whether the respondent commutes by bicycle, yes = 1, no = 0		0.04	0.20	0.00	1.00
Independent variable	Built environment	District Location	Distance to the urban center	Distance to Hankou, the first-class urban center of Wuhan (in km)	Origin	12.32	13.28	0.13	77.54
					Destination	12.12	13.24	0.03	78.05
			Distance to the nearest cluster center	Distance to the center of the nearest cluster (in km)	Origin	7.91	10.91	0.16	76.68
					Destination	8.47	12.11	0.01	76.68
		Public transport accessibility	Distance to the nearest public transit station	Distance (in km) from the respondent’s residence to the nearest bus stop (both metro and surface bus)	Origin	1653.56	6360.54	2.49	48,897.77
					Destination	1.44	5.59	0.00	45.28
			Public transit station number	Number of public transit stops within a 15 min walking isochronous circle of the respondent	Origin	17.07	10.42	0.00	46.00
					Destination	17.13	10.37	0.00	53.00
		Density	Population density	Residential density (persons/km^2^) within a 15 min walking isochronous circle of respondents	Origin	24,563.92	17,631.81	37.65	75,166.52
					Destination	23,988.06	17,393.03	0.00	75,493.31
			Job density	Job density (persons/km^2^) within a 15 min walking isochronous circle of respondents	Origin	8212.74	5746.78	43.22	24,359.55
					Destination	8549.08	5954.31	32.80	24,197.97
			Land use intensity	Floor area ratio of sites within a 15 min walking isochronous circle of the respondent	Origin	2.86	1.61	0.00	6.25
					Destination	2.80	1.68	0.00	6.25
		Design	Intersection density	Density of intersections of four or more roads within a 15 min walking isochronous circle of respondents (pcs/km^2^)	Origin	30.19	20.89	0.00	119.00
					Destination	30.01	20.59	0.00	123.00
			Road network density	Density of the road network within a 15 min walking isochronous circle of the respondent (in km/km^2^)	Origin	6010.36	2202.65	81.17	13,373.12
					Destination	5977.54	2196.62	0.00	13,515.15
		Diversity	Land use entropy index	Mixed entropy of land use within a 15 min walk isochronous circle of respondents	Origin	0.64	0.13	0.00	0.98
					Destination	0.65	0.13	0.00	0.99
	Socio-demographics		Age	Age of respondent		31.53	12.70	6.00	75.00
			Gender	Respondent gender, dummy variable, male = 1, female = 0		0.50	0.50	0.00	1.00
			Occupation	Whether the respondent is a full-time working employee, dummy variable, yes = 1, no = 0		0.62	1.08	0.00	1.00
			Family number	Number of family members interviewed		2.89	0.93	1.00	6.00
			Family income	Respondent’s annual household income, dummy variable, less than 50,000¥ = 1, 50,000–100,000¥ = 2, 100,000–250,000¥ = 3, 250,000–400,000¥ = 4, 400,000–550,000¥ = 5, 550,000–700,000¥ = 6, greater than 700,000¥ = 7		2.85	0.83	1.00	7.00
			Car ownership	Whether the respondent’s household owns a private car, dummy variable, yes = 1, no = 0		0.56	1.08	0.00	1.00
			House size	Respondent’s household housing size, dummy variable, below 40 m^2^ = 1, 40–70 m^2^ = 2, 70–90 m^2^ = 3, 90–110 m^2^ = 4, 110–120 m^2^ = 5, 120–150 m^2^ = 6, greater than 150 m^2^ = 7		3.58	1.14	1.00	7.00
			Education level	Respondents’ degree status, dummy variable, primary school and below = 1, middle school = 2, high school = 3, undergraduate = 4, undergraduate and above = 5, postgraduate and above = 6		3.14	1.08	1.00	6.00

**Table 5 ijerph-20-01969-t005:** Model parameters table.

Model	Model 1	Model 2
Best decision tree scale n	35	141
Best split variable number m	6	6
Best tree depth d	46	24
R^2^	0.77	0.80

**Table 6 ijerph-20-01969-t006:** Relative contributions of independent variables on active mode choice.

		Walk			Bike		
Categories	Predictor Variables	Ranking	Relative Importance		Ranking	Relative Importance	
Individual Attributes	Age	1	6.65%	16.78%	3	4.74%	14.70%
	Gender	26	1.05%		27	0.87%	
	Education level	22	2.28%		22	1.98%	
	Housing area	23	2.05%		24	1.69%	
	Employment status	28	0.55%		23	1.82%	
	Car ownership	27	0.68%		28	0.72%	
	Family income	25	1.60%		26	1.33%	
	Family number	24	1.92%		25	1.55%	
Built environment attributes at residential location	o_ Distance to the city center	6	4.52%	41.13%	7	4.67%	42.92%
	o_ Land use intensity	13	4.12%		13	4.35%	
	o_ Intersection density	20	3.14%		19	3.59%	
	o_ Land use entropy index	10	4.36%		5	4.68%	
	o_ Distance to the nearest cluster center	2	5.56%		14	4.32%	
	o_ Population density	16	4.01%		9	4.64%	
	o_ Road network density	17	3.78%		6	4.68%	
	o_ Number of public transit stations	19	3.21%		20	3.17%	
	o_ Distance to the nearest stop	14	4.04%		16	3.99%	
	o_ Job density	9	4.39%		2	4.84%	
Built environment attributes at work location	d_ Distance to the city center	3	4.77%	42.09%	4	4.71%	42.38%
	d_ Land use intensity	12	4.18%		17	3.96%	
	d_ Intersection density	18	3.53%		18	3.59%	
	d_ Land use entropy index	8	4.40%		11	4.49%	
	d_ Distance to the nearest cluster center	4	4.67%		10	4.53%	
	d_ Population density	15	4.03%		1	5.04%	
	d_ Road network density	5	4.65%		12	4.44%	
	d_ Number of public transit stations	21	3.11%		21	2.96%	
	d_ Distance to the nearest stop	7	4.41%		15	4.00%	
	d_ Job density	11	4.33%		8	4.66%	

The prefix “d_” means the built environment variable of the respondent’s destination. The prefix “o_” means the built environment variable of the respondent’s residence.

## Data Availability

Not applicable, because the used data is confidential.

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
