# Peer review of "Examining the Nonlinear Effects of Residential and Workplace-built Environments on Active Travel in Short-Distance: A Random Forest Approach"

_ijerph, 2023, doi:10.3390/ijerph20031969_

Round 1

Reviewer 1 Report

As the number of cars increases annually, environmental pollution and health problems caused by excessive car use are of global concern. Therefore, it is necessary to discuss how can we promote people's choices of replacing some car trips for short trips with active travel by walking or cycling. Although the paper is overall well written and much of it to be well described, I still find some descriptions of very important points are inadequate or completely missing. Therefore, I recommend that a major revision is warranted. I explain my concerns in more detail below:

1Literature review is not well written. The research gaps in lines 47-48, 50-51, 53-55 and 101-102 cannot be concluded from literature review.

2The number of survey population in lines163-164 does not correspond to Table 1.

3Figures are so blurred in the manuscript, especially figure 4,5,6 and 7, that I can’t make a judgment about the threshold-related conclusions.

4Results numbers could be much more precise. A lot of mistakes have been found.

1)For example, lines 293-294authors have mentioned that the three most influential factors on cycling were: job density (4.83%), land-use diversity (4.67%), and road network density (4.67%). But as shown in Table6, the relative importance of land-use diversity is 4.68%. Since the variables by more than 0.2% are defined to differ relatively more(lines304-305), more precise results are encouraged. There are many similar mistakes in the manuscript.

2)For example, lines 265-267, the total contribution of the 10 built environments of residence to the influence role of walking and cycling was 41.69% and 44.75%. But as shown in Table6, the correct numbers are 41.13% and 42.93%respectively. There are many similar mistakes in the manuscript.

3)Line 277-278it is corrected that age is followed by education level and housing area.

5A more condensed result and a more insightful discussion are strongly suggested. Lines 125-141 are well written. Just as the ideas of lines 125-141, it is encouraged to combine the real situation in Wuhan with the results in the manuscript to propose more in-depth discussions and applicable suggestions.

Reviewer 2 Report

In this paper, a random forest approach was used to identify the non-linear effects of the built environment of the residence and workplace on two active travel modes, walking and cycling. While the relative importance of different factors was also assessed. This study may be helpful to improve the urban transportation travel structure and urban environment. Overall, this paper is well constructed and logical rigour. The topic is worthy of investigation and is of interest to the readers of the journal. Hereafter are some comments which may make the paper clearer and easier to read.

1.    The clarity of the pictures in the text needs special attention, the axes of the pictures are currently not obvious and are difficult to read (Fig.4 ~ Fig.7).

2.    The RF model selected 20% of the samples randomly selected as the test set and 80% as the training set, in order to obtain more reliable model results.” Is the validation of the resulting model based on a test dataset and were the 20% selection criteria justified? This section needs further clarification.

3.    The findings of this study differ from those of previous studies in Western countries. Where are the differences specific? Is this linked to the national context and policy? More discussions should be provided about this issue.

Round 2

Reviewer 1 Report

I would thank the effort the authors spend on resolving the comments. The majority of my comments have been successfully resolved and properly responded. But I still have three suggestions for the authors.

(1) The pictures are currently blurred and difficult to read (Fig.1 and Fig.3).  

(2) Results are not well written. The function of figures is to support results. The results that 5 of the 10 built environment factors of residential location had a threshold effect on the choice of walking, and all ten residential-built environment factors had a threshold effect on the choice of cycling can't be achieved from Figure 4-5. Similar problem still exists in 4.3.  

(3) Results need to be addressed clearly in abstract. 
